# Systematic Review and Meta-Analysis on the Sensitivity and Specificity of ^13^C/^14^C-Urea Breath Tests in the Diagnosis of *Helicobacter pylori* Infection

**DOI:** 10.3390/diagnostics12102428

**Published:** 2022-10-07

**Authors:** Layal K. Jambi

**Affiliations:** Radiological Sciences Department, College of Applied Medical Sciences, King Saud University, Riyadh 11433, Saudi Arabia; ljambi@ksu.edu.sa; Tel.: +966-55-310-1005

**Keywords:** *Helicobacter pylori*, *H. pylori*, meta-analysis, sensitivity, specificity, urea breath tests

## Abstract

*Helicobacter pylori* (*H. pylori*) continues to be a major health problem worldwide, causing considerable morbidity and mortality due to peptic ulcer disease and gastric cancer. The aim of the present systematic review and meta-analysis was to determine the sensitivity and specificity of ^13^C/^14^C-urea breath tests in the diagnosis of *H. pylori* infection. A PRISMA systematic search appraisal and meta-analysis were conducted. A systematic literature search of PubMed, Web of Science, EMBASE, Scopus, and Google Scholar was conducted up to August 2022. Generic, methodological and statistical data were extracted from the eligible studies, which reported the sensitivity and specificity of ^13^C/^14^C-urea breath tests in the diagnosis of *H. pylori* infection. A random effect meta-analysis was conducted on crude sensitivity and specificity of ^13^C/^14^C-urea breath test rates. Heterogeneity was assessed by Cochran’s Q and ^I2^ tests. The literature search yielded a total of 5267 studies. Of them, 41 articles were included in the final analysis, with a sample size ranging from 50 to 21857. The sensitivity and specificity of ^13^C/^14^C-urea breath tests in the diagnosis of *H. pylori* infection ranged between 64–100% and 60.5–100%, respectively. The current meta-analysis showed that the sensitivity points of estimate were 92.5% and 87.6%, according to the fixed and random models, respectively. In addition, the specificity points of estimate were 89.9% and 84.8%, according to the fixed and random models, respectively. There was high heterogeneity among the studies (I^2^ = 98.128 and 98.516 for the sensitivity and specificity, respectively, *p*-value < 0.001). The ^13^C/^14^C-urea breath tests are highly sensitive and specific for the diagnosis of *H. pylori* infection.

## 1. Introduction

The spiral-shaped bacteria *Helicobacter pylori* (*H. pylori*) can be found attached to the stomach’s epithelial lining or in the gastric mucous layer [1]. *H. pylori* continues to be a serious health issue in the world, contributing significantly to morbidity and mortality from stomach cancer and peptic ulcer disease [2]. More than 90% of duodenal ulcers and up to 80% of stomach ulcers are brought on by *H. pylori* [3]. Recent research has demonstrated a link between chronic *H. pylori* infection and the emergence of stomach cancer [4]. Additionally, *H. pylori* infection affects almost two-thirds of the world’s population [5]. According to earlier estimates, more than 50% of people worldwide have *H. pylori* in their upper gastrointestinal tracts [6,7], with developing nations having a higher prevalence of this illness. In addition, in countries with poor sanitation, 90% of the adult population can be affected [8].

Most *H. pylori*-infected people never experience any signs of the infection, whereas *H. pylori* causes chronic active, chronic persistent, and atrophic gastritis in both adults and children [9]. Furthermore, contamination with *H. pylori* causes gastric and duodenal ulcers [10]. Additionally, the infection has been linked to the emergence of several malignancies [11]. Comparatively to those who are not infected with *H. pylori*, infected individuals have a 2- to 6-fold greater risk of developing gastric cancer and mucosal-associated lymphoid-type (MALT) lymphoma [12]. Additionally, extranodal marginal zone B-cell lymphoma of the listed organs (stomach, esophagus, colon, rectum, or tissues around the eye) and diffuse large B-cell lymphoma (stomach) lymphoid tissue cancers have also been linked to *H. pylori* [13,14].

Researchers have hypothesized that *H. pylori* influences or protects against a variety of different disorders, but many of these connections are still debatable [15]. According to certain research, *H. pylori may* play a significant part in the natural ecology of the stomach, such as by affecting the types of bacteria that colonize the gastrointestinal tract [16]. Other research points to the potential benefits of non-pathogenic *H. pylori* strains in normalizing stomach acid output and controlling appetite [17]. People who have active duodenal or gastric ulcers or a known history of ulcers should be tested for *H. pylori* and should be treated if discovered to be infected. Following resection of early gastric cancer and for low-grade gastric MALT lymphoma, screening for and managing *H. pylori* infection is advised [18].

Invasive and non-invasive testing techniques can also be used to diagnose *H. pylori* infection [19]. An invasive method to check for *H. pylori* infection is an endoscopic biopsy. The best way to identify *H. pylori* infection is to combine a fast urease test or microbial culture with a histological study of two sites following an endoscopic biopsy [20]. The carbon urea breath test, stool antigen testing, and blood antibody tests are a few non-invasive tests for *H. pylori* infection that may be appropriate [18].

The ^13^C and ^14^C tests are two urea breath test (UBTs) that have been approved by the Food and Drug Administration. Both exams are reasonably priced and provide real-time results availability. Although the radiation dose is relatively low (approximately 1 microCi) [21] and the ^14^C-UBT uses a radioactive isotope, some doctors may prefer the ^13^C test since it is non-radioactive compared to ^14^C, especially in young children and pregnant women. ^14^C-UBT has a high degree of diagnostic accuracy [22].

However, it should be noted that the UBT has occasionally failed to provide an accurate diagnosis because its accuracy has frequently been compared to stomach biopsies, the gold standard, and it is widely recognized that the latter test is prone to sampling error (because of the discontinuous *H. pylori* colonization of the gastric mucosa). For instance, the low specificity (e.g., high rate of false positives) of the UBT in some of the comparison studies may actually be attributable to the test’s low sensitivity [23]. This has been confirmed by some writers who discovered that numerous UBT results that appeared to be false positives were actually correct results, as was amply demonstrated by the examination of additional multiple biopsy specimens taken from the same patients who underwent a second gastroscopy [24].

To confirm *H. pylori* colonization and to track its eradication, UBT is advised. A positive UBT results in an active *H. pylori* infection, necessitating treatment or additional testing using invasive techniques. Initial *H. pylori* treatment involves either triple, quadruple, or sequential therapy regimens, all of which contain a proton pump inhibitor in addition to a variety of antibiotics; treatment durations typically range from 7 to 14 days [25].

The patient is given either ^13^C or ^14^C-labeled urea to drink during this test. Rapid urea metabolization by *H. pylori* results in absorption of the tagged carbon. If *H. pylori* is present, the tagged carbon can subsequently be quantified as carbon dioxide in the patient’s exhaled breath [26].

Numerous changes have been suggested since Graham et al.’s initial description of the 13C-UBT to precisely detect *H. pylori* infection [27]. Changes have been made to the test meal type, the period of breath collection, the cut-off values, and the equipment used to determine isotope enrichment, among other things. A definitive standardization of this test has not yet been established, despite the fact that many variations to the UBT approach have been suggested and a large number of studies pertaining to its methodology have been published. Hence, the current systematic review aims to determine the sensitivity and specificity of ^13^C/^14^C-urea breath tests in the diagnosis of *H. pylori* infection.

## 2. Materials and Methods

### 2.1. PRISMA Guidelines and Protocol Registration

This systematic review and meta-analysis were produced following the Preferred Reporting Items for Systematic Reviews and Meta-Analyses (PRISMA) guidelines (Appendix A). The study protocol was registered at The International Prospective Register of Systematic Reviews (PROSPERO, registration No. CRD42022354308).

### 2.2. Literature Search Strategy

The author systematically and comprehensively searched English articles published in PubMed, Web of Science, EMBASE, Scopus, and Google Scholar without a region or time limit. To increase the search scale, a combination of search strategies was carried out; First: search MESH (medical subject heading) using the following terms: “^13^C UBT”, “^14^C UBT”, and “*Helicobacter Pylori* Infection” (Appendix A); Second free-text search using the following search keywords: “*Helicobacter pylori*”, “*H. pylori*”, “Helicobacter infection”, “dyspepsia”, “gastritis”, “urea breath test”, “breath test”, “^13^C-UBT”, “^14^C-UBT”, “UBT”, “Sensitivity”, and “Specificity”. Boolean operator (OR) was used to combine synonyms, and (AND) was used to combine the cases with tests.

### 2.3. Eligibility Criteria (Inclusion/Exclusion)

Articles studied the sensitivity and specificity of ^13^C/^14^C-urea breath tests in the diagnosis of *H. pylori* infection were included in this systematic review and meta-analysis. However, articles where the end measure was not the sensitivity and specificity of ^13^C/^14^C-urea breath tests in the diagnosis of *H. pylori* infection, review articles, case report articles, non-English language articles, articles lacking pertinent data, and articles without full text, were excluded.

### 2.4. Study Screening

The EndNote V.X8 software was employed for the article screening process management. The duplicates were omitted, then the author meticulously selected the included articles by screening the titles, abstracts, and full texts of the articles.

### 2.5. Data Extraction

The required data were extracted in a standardized table that included the following headings: name of the first author and year of publication, country of the study, the population of the study, the sample size of the study, mean age of the study participants, gender of the study participants, study design, name of UBT (^13^C or ^14^C), the dose of ^13^C/^14^C -urea, using of infrared, the cut-off UBT threshold, the used gold standard for *H. pylori* detection, the duration from ingestion of carbon isotope to the collection of post-ingestion exhaled air, diagnostic sensitivity percentages, and diagnostic specificity percentages.

### 2.6. Quality Assessment

The Quality Assessment Tool for Quantitative Studies (QATFQS), established by the Effective Public Health Practice Project (EPHPP) [28], was used to assess the quality of the included studies. Each article underwent eight items of the tool, which were individually scored as “1” denotes strong, “2” denotes moderate, and “3” denotes poor quality. Thereafter, the overall rating for each study was calculated based on the following criteria: “1” denotes strong (no ratings of weak), “2” denotes moderate (one rating of weak), and “3” denotes weak quality (two or more weak ratings) [29].

### 2.7. Meta-Analysis

The meta-analysis was achieved by using Comprehensive Meta-Analysis Software (CMA, version 3, BioStat, Tampa, FL, USA). The Fail-Safe N test was employed to calculate the effect size values of the included studies.

Publication bias was identified by the funnel plot, and Begg’s and Mazumdar’s rank correlation tests were used to detect the potential publication bias among the included articles. Kendall’s tau is used to understand the strength of the relationship between two variables.

The I-squared (I^2^) statistic was used to measure the heterogeneity between the included articles, and values of 25%, 50%, and 75% are regarded as low, moderate, and high estimations, respectively [30]. A non-significant level of statistical heterogeneity is assumed when the *p*-value > 0.05. The implementation of a random-effect model was prompted by the substantial level of variability [31]. 

The Meta-Disc 1.4 software was used in the analysis of the likelihood ratio for positive and negative test (LR+ and LR-), and symmetric receiver operating characteristics (SROC) curve.

## 3. Results

### 3.1. Search Findings

A total of 5,267 articles were obtained from the search in the five databases: PubMed (n = 1756), Web of Science (n = 512), EMBASE (n = 187), Scopus (n = 867), and Google Scholar (n = 1945). The remained number of articles after omitting the duplicates was 2377. Then, 1363 and 1015 articles were removed by title and abstract screening, respectively. The full texts of the remained 248 articles were assessed. Finally, 207 articles were excluded for not meeting the inclusion criteria. The screening process resulted in 41 articles being accepted for qualitative synthesis and meta-analysis (Figure 1).

### 3.2. Characteristics of the Included Articles

The majority of the included articles were published in 2003 (six), followed by 2000 (four), 2002 (four), 1998 (two), 2001 (two), 2005 (two), 2007 (two), 2009 (two), 2021 (two), and one for 1989, 1991, 1996, 1997, 1999, 2004, 2008, 2011, 2012, 2015, 2017, 2018, and 2019. The majority of the articles were conducted in China (seven), followed by Japan (three), Taiwan (three), Spain (three), Turkey (three), the United States (two), Italy (two), and one from Germany, Australia, Brazil, Belgium, Poland, Mexico, Estonia, India, Jordan, Iraq, Egypt, Israel, South Korea, Indonesia, New Zealand, Singapore, Pakistan, and Austria. The vast majority of the articles were cross-sectional in design. Only one study was performed on children, and the rest recruited adults. The sample size of the articles ranged between 50 and 21857 participants. Out of the 41 included articles, only 12 articles reported the sensitivity and specificity of ^14^C-urea breath tests in the diagnosis of *H. pylori* infection. Obtained breath samples were analyzed using an isotope-selective, non-dispersive infrared spectrometer in only nine articles. The duration from ingestion of carbon isotope to the collection of post-ingestion exhaled air ranged between 5–30 minutes. The sensitivity and specificity of ^13^C/^14^C-urea breath tests in the diagnosis of *H. pylori* infection ranged between 64–100% and 60.5–100%, respectively (Table 1).

### 3.3. Unified Findings

The effect analysis of the 41 included articles ii the current meta-analysis showed that the sensitivity points of estimate were 0.925 and 0.876 according to the fixed and random models, respectively. In addition, the specificity points of estimate were 0.899 and 0.848 according to the fixed and random models, respectively. The Q value calculated by the homogeneity test shows that sensitivity and specificity of ^13^C/^14^C-urea breath tests in the diagnosis of *H. pylori* infection data have a heterogenous structure (Q = 2137.239; *p* < 0.001) and (Q = 2689.815; *p* < 0.001). Accordingly, the author achieved the current meta-analysis following the random-effects model to reduce the misapprehensions produced by the article’s heterogenicity. The total actual heterogeneity between the included articles (tau value) was 1.830 and 1.809 for sensitivity and specificity, respectively.

The obtained high level of heterogeneity, I-squared (I^2^), 98.128, and 98.516 for the sensitivity and specificity, respectively, indicating that the random effect model for meta-analysis should be applied (Table 2).

### 3.4. The Sensitivity and Specificity of ^13^C/^14^C-Urea Breath Tests in the Diagnosis of Helicobacter pylori Infection

Even though the random-effect meta-analysis showed a high heterogeneity among articles, I^2^ = 98.128, the pooled event rates and (95%CIs) for the sensitivity and specificity were 0.876 (95%CI: 0.799–0.926) and 0.848 (95%CI: 0.760–0.908), respectively (Figure 2) and (Figure 3).

Results indicated that the ^13^C/^14^C-urea breath tests have very high sensitivity and specificity in the diagnosis of *Helicobacter pylori* infection. The pooled urea breath test result. overall likelihood ratio for positive test, overall likelihood ratio for negative test and the symmetric receiver operating characteristics (SROC) curve are presented in Figure 4, Figure 5, and Figure 6, respectively.

### 3.5. Fail-Safe N Method

The Fail-Safe N was employed to assess the robustness of a significant result by calculating how many studies with effect size zero could be added to the meta-analysis before the result lost statistical significance. The Z-values for observed studies for sensitivity and specificity were 44.48086 and 40.05841, respectively, indicating that the effect value achieved by our meta-analysis is susceptible to publication bias (Table 3).

### 3.6. Rank Correlation

Begg and Mazumdar test showed a moderate relationship between the sensitivity of ^13^C/^14^C-urea breath tests in the diagnosis of *H. pylori* infection, tau with and without continuity correction; values were 0.28729 and 0.28851, respectively. Furthermore, a strong relationship between the sensitivity of ^13^C/^14^C-urea breath tests in the diagnosis of *H. pylori* infection, tau with and without continuity correction values were 0.39560 and 0.39683, respectively. Egger’s test for a regression intercept provided *p*-values of 0.02528 and 0.02771 for sensitivity and specificity, respectively, indicating the presence of publication bias (Table 4).

### 3.7. Publication Bias

The asymmetric funnel plots of the sensitivity and specificity of ^13^C/^14^C-urea breath tests in the diagnosis of *H. pylori* infection suggest publication bias (Figure 7).

The subgroup analysis showed that both ^13^C/^14^C-urea breath tests have high sensitivity and specificity in the diagnosis of *H. pylori* infection without a significant difference (Table 5).

## 4. Discussion

*H. pylori* continues to be a serious health issue in the world, contributing significantly to morbidity and mortality from stomach cancer and peptic ulcer disease [1]. Testing for *H. pylori* is advised in some cases of dyspepsia, after endoscopic resection of early gastric cancer, for first-degree relatives with gastric cancer, and in the presence of peptic ulcer disease or low-grade gastric MALT lymphoma [2]. There are numerous testing techniques, both invasive and non-invasive. Endoscopy is necessary for invasive techniques including histology, culture, and fast urease in order to take stomach mucosa biopsies. Despite the excellent specificity of these tests, their sensitivity may be compromised by the localized localization of the infection in the stomach [73]. Additionally, endoscopy might be too resource-intensive and lab facilities might not be able to culture the organism in the under-developed countries, where *H. pylori* are most common [74].

The carbon urea breath test, stool antigen testing, and blood antibody tests are a few non-invasive tests for *H. pylori* infection that may be appropriate [3]. The spiral bacterium *H. pylori*, which has been linked to gastritis, stomach ulcers, and peptic ulcer disease, can be quickly diagnosed by the urea breath test [4]. It is predicated on *H. pylori’s* capacity to transform urea into ammonia and carbon dioxide. Leading society guidelines support urea breath tests as the primary non-invasive option for identifying *H. pylori* both before and after treatment [5]. Patients in this study ingest urea that has been labeled with either radioactive carbon-14 or non-radioactive carbon-13, a rare isotope. The detection of isotope-labeled carbon dioxide in the exhaled breath during the following 10 to 30 minutes reveals that the urea was divided; this reveals the presence of urease in the stomach and, consequently, the presence of *H. pylori* bacteria [6]. The urea breath test has some limitations because it only detects active *H. pylori* infections; as a result, it will detect less urease if antibiotics are reducing the number of *H. pylori* present, or if the stomach’s acidity is lower than usual. The test should only be carried out 14 days after ceasing the use of the acid-reducing medicine, such as proton pump inhibitors, or 28 days after ceasing antibiotic therapy. Additionally, a reservoir of *H. pylori* in dental plaque, according to some physicians, may also have an impact on the outcome of the urea breath test [7].

Determining the sensitivity and specificity of ^13^C/^14^C-urea breath tests in the detection of *H. pylori* infection was the purpose of the current systematic review and meta-analysis. In all, 5267 studies were found in the literature search for the current meta-analysis. A sample size ranging from 50 to 21857 articles from among them that met the inclusion criteria were included in the final analysis. The primary findings of the present meta-analysis demonstrated that the sensitivity and specificity of ^13^C/^14^C-urea breath tests in the detection of *H. pylori* infection were between 64 and 100% and 60.5 and 100%, respectively. Further analysis revealed that the fixed and random models’ respective sensitivity points of estimation were 92.5% and 87.6%. The specificity points of estimate for the fixed and random models were 89.9% and 84.8%, respectively.

We showed in this meta-analysis that the ^13^C/^14^C-urea breath tests are very sensitive and specific for identifying *H. pylori* infection. In addition to being non-invasive, the urea breath test has the benefit of offering a thorough evaluation independent of the potential sampling error associated with endoscopic biopsy. Other drawbacks of the biopsy tests include their reliance on the pathologist’s expertise and experience, as well as studies that have shown intern observer variability [8]. The diagnosis of *H. pylori* infection can be made using a variety of invasive and non-invasive techniques, including endoscopy with biopsy, immunoglobulin titer serology, stool antigen analysis, and the UBT. The non-invasive, user-friendly characteristics of UBT make it possible that this detection technology will be adopted in many therapeutic situations. In the diagnostic assessment of dyspeptic patients with comorbidities that enhance their risk of upper endoscopy, who are unable to undergo upper endoscopy, or who have known or suspected stomach atrophy, UBT can be helpful. The findings of the study imply that UBT has a good diagnostic specificity for identifying *H. pylori* infection in dyspeptic patients. However, the UBT is an expensive procedure that necessitates specialized carbon dioxide monitoring equipment and radioactive material management infrastructure [9].

The current meta-analysis also revealed that there was significant heterogeneity among studies (I^2^ = 98.128 and 98.516 for the sensitivity and specificity, respectively, *p*-value 0.001), which may be attributed to the use of various types of reference standards, timing between capsule consumption and testing, or differences in the methodological quality of the included studies. The nature of the radioisotope meal and specific patient characteristics may also affect within-study variability, in addition to the urease activity of the oral flora which can affect the reading of the urea breath test, the cut-off value and the time to take the reading after the meal ingestion which were not clearly stated in many of the studies involved. However, despite the fact that our research was unable to identify this difference, it is quite likely that test performance varies across individuals with varied pre-test risks.

Despite earlier meta-analyses discussing the sensitivity and specificity of ^13^C/^14^C-urea breath tests in the diagnosis of *H. pylori* infection, this study provides updated evidence-based information using a comprehensive search of electronic databases for pertinent publications. The significant drawbacks were the considerable heterogeneity that remained unexplained despite several subgroup analyses, and the fact that only papers published in English were included. In addition, this meta-analysis calculated the sensitivity and specificity of both ^13^C/^14^C-urea breath tests in the diagnosis of *H. pylori* infection, therefore, separated analysis for ^13^C and ^14^C-urea breath tests is recommended. Adults and children who had non-invasive tests to diagnose *H. pylori* were included in this systematic review and meta-analysis. The majority of the articles only enrolled symptomatic participants, hence the conclusions of this study only apply to those who have symptoms. Most studies did not include individuals with prior gastrectomy, recent antibiotic or proton pump inhibitor use, or recent gastrectomy. The results of this study therefore do not apply to these populations.

## 5. Conclusions

For the diagnosis of *H. pylori* infection, the ^13^C/^14^C-urea breath tests are extremely sensitive and specific. The results of this investigation should therefore be regarded with caution because considerable heterogeneity also limits the usefulness of diagnostic meta-analytic estimates. In addition, this meta-analysis calculated sensitivity and specificity of both ^13^C/^14^C-urea breath tests in the diagnosis of *H. pylori* infection, therefore, separated analysis for ^13^C and ^14^C-urea breath tests is recommended.

## Figures and Tables

**Figure 1 diagnostics-12-02428-f001:**
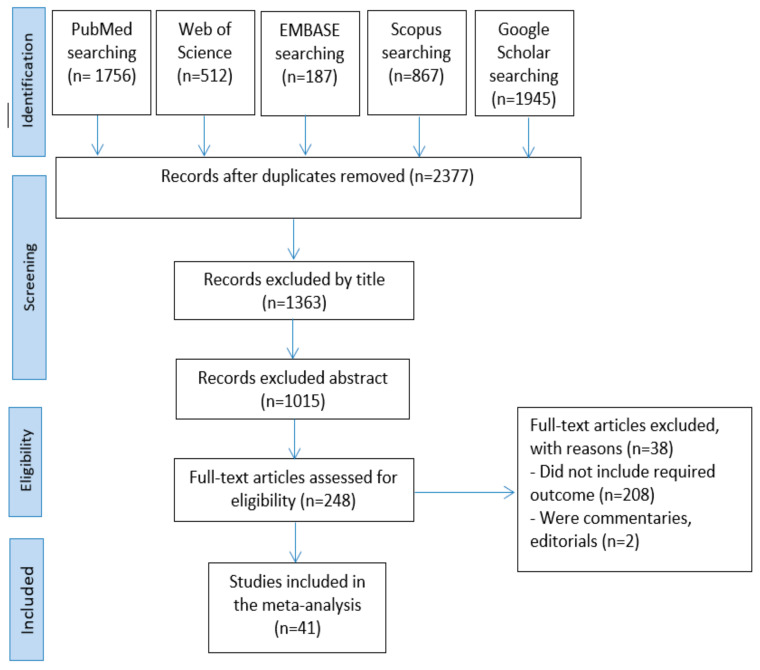
Articles selection process.

**Figure 2 diagnostics-12-02428-f002:**
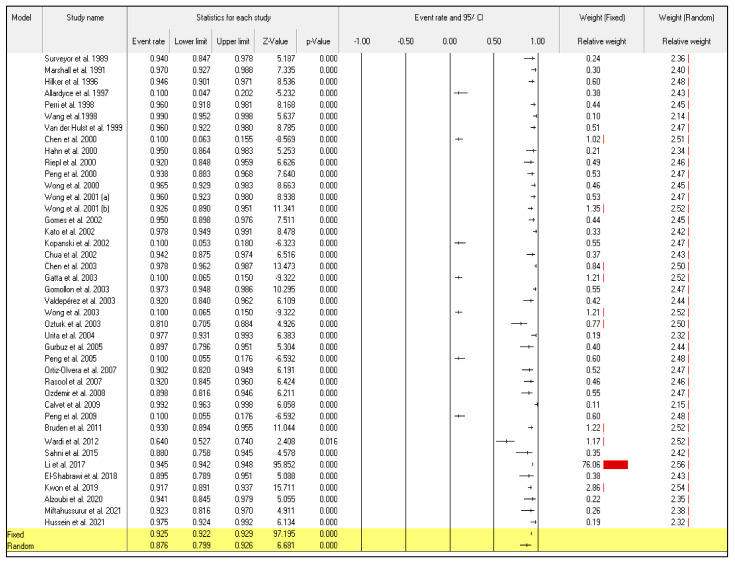
Forest plot from the fixed and random-effects analysis: the sensitivity of ^13^C/^14^C-urea breath tests in the diagnosis of *Helicobacter pylori* infection [32,33,34,35,36,37,38,39,40,41,42,43,44,45,46,47,48,49,50,51,52,53,54,55,56,57,58,59,60,61,62,63,64,65,66,67,68,69,70,71,72].

**Figure 3 diagnostics-12-02428-f003:**
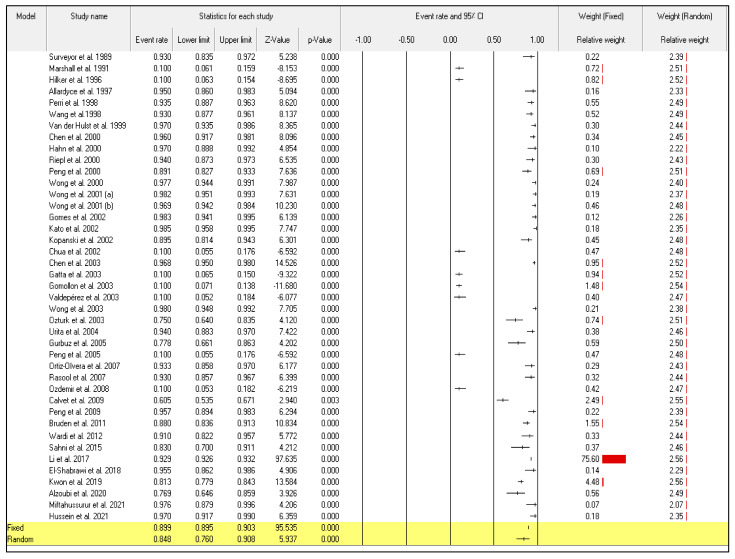
Forest plot from the fixed and random-effects analysis: the specificity of ^13^C/^14^C-urea breath tests in the diagnosis of *Helicobacter pylori* infection [32,33,34,35,36,37,38,39,40,41,42,43,44,45,46,47,48,49,50,51,52,53,54,55,56,57,58,59,60,61,62,63,64,65,66,67,68,69,70,71,72].

**Figure 4 diagnostics-12-02428-f004:**
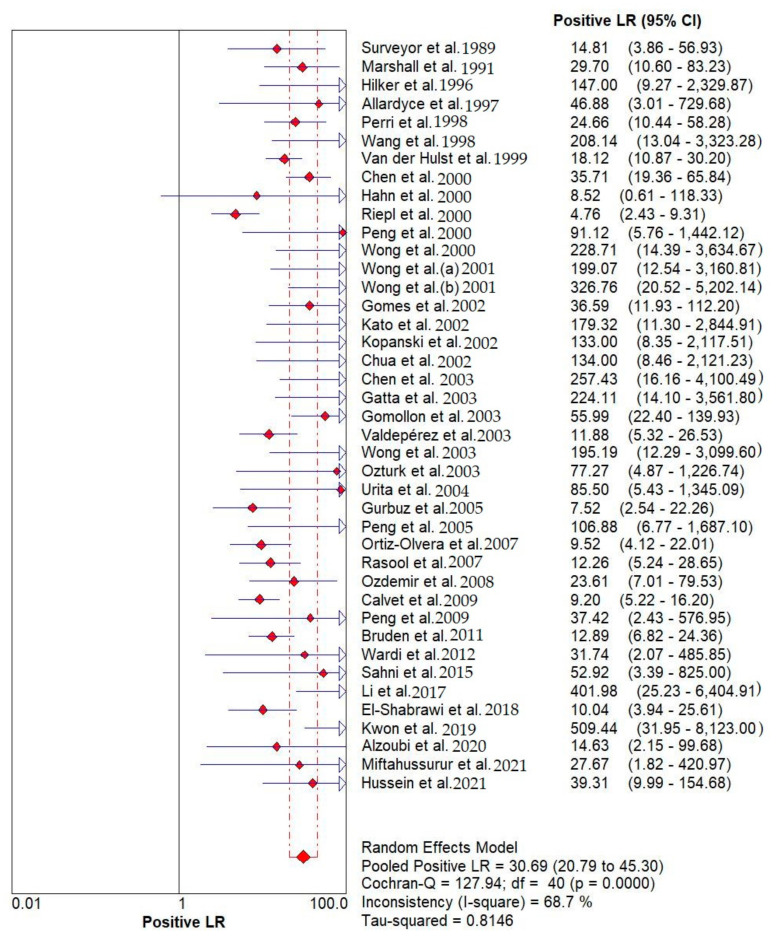
Pooled urea breath test result. Overall likelihood ratio for positive test [32,33,34,35,36,37,38,39,40,41,42,43,44,45,46,47,48,49,50,51,52,53,54,55,56,57,58,59,60,61,62,63,64,65,66,67,68,69,70,71,72].

**Figure 5 diagnostics-12-02428-f005:**
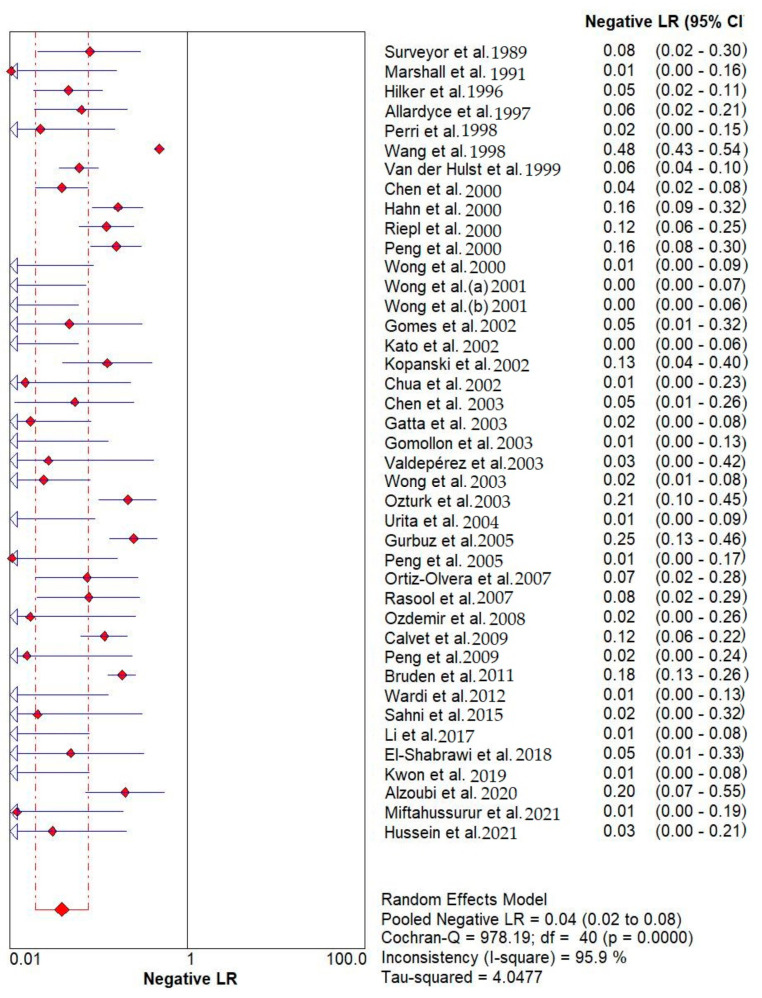
Pooled urea breath test result. Overall likelihood ratio for negative test [32,33,34,35,36,37,38,39,40,41,42,43,44,45,46,47,48,49,50,51,52,53,54,55,56,57,58,59,60,61,62,63,64,65,66,67,68,69,70,71,72].

**Figure 6 diagnostics-12-02428-f006:**
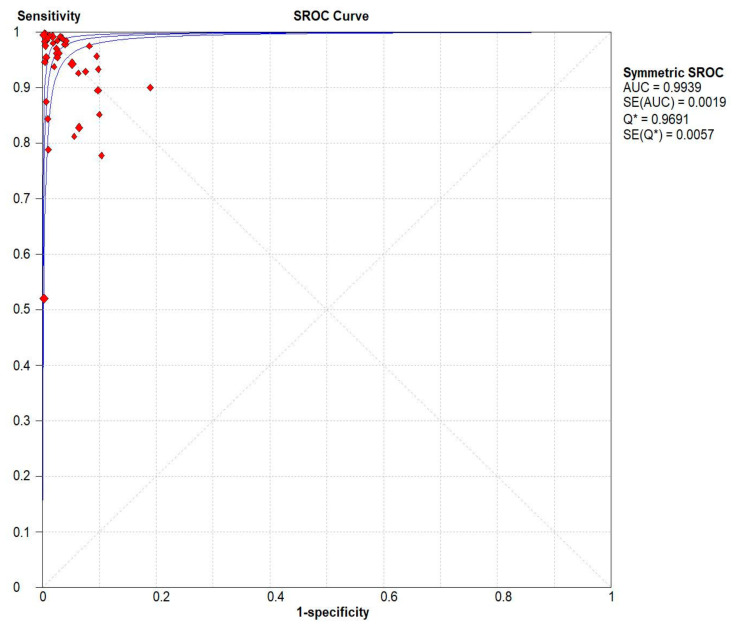
Pooled urea breath test result. Symmetric receiver operating characteristics (SROC) curve.

**Figure 7 diagnostics-12-02428-f007:**
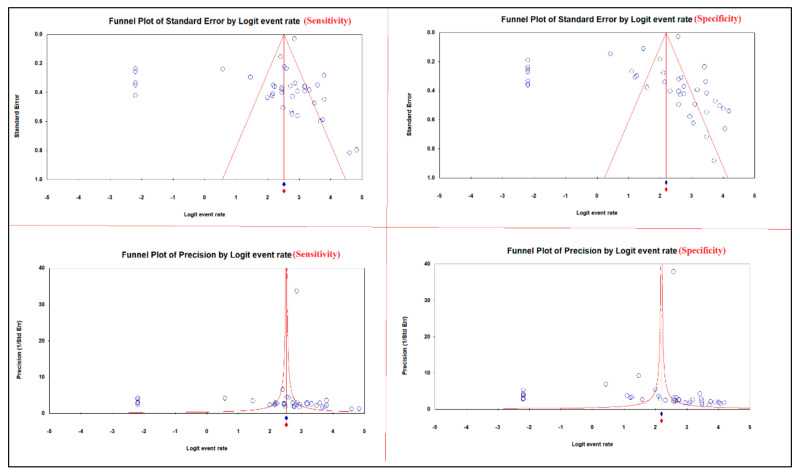
Publication bias of the sensitivity and specificity of ^13^C/^14^C-urea breath tests in the diagnosis of *Helicobacter pylori* infection.

**Table 1 diagnostics-12-02428-t001:** The extracted data from the included articles.

The First Author (Year)	Country	Study Population	Sample Size	Mean Age (Year)	Gender (M:F)	Study Design	Name of UBT (^13^C/^14^C)	Dose of ^13^C/^14^C-Urea	Infrared Assisted	Cut-Off UBT Threshold	Reference Standard	Time	Sensitivity (%)	Specificity (%)
Surveyor et al. 1989 [32]	Australia	Patients underwent upper GI endoscopy	63	58.8	33:30	Cross-sectional	^14^C	NM	No	NM	Histo and/or culture	Every 5 min for 30 min	94.0	93.0
Marshall et al. 1991 [33]	United States	Patients underwent gastroscopy with biopsy	153	NM	NM	Cross-sectional	^14^C	NM	No	>6%	Combined (Culture, RUT and histo)	30 min after administration	97.0	100.0
Hilker et al. 1996 [34]	Germany	Patients underwent urea breath test and gastroscopy with biopsy	174	46	68:106	Retrospective cross-sectional	^13^C	NM	No	>250	Histo	30 min after administration	94.6	100.0
Allardyce et al. 1997 [35]	New Zealand	Patients with dyspeptic symptoms	63	56.5	37:26	Cross-sectional	^14^C	37	No	82% DPM	Histo or (Biopsy and rapid urea test)	30 min and 60 min post ingestion	100.0	95.0
Perri et al. 1998 [36]	Belgium	Patients underwent upper GI endoscopy	172	39.7	14:158	Cross-sectional	^13^C	NM	No	3.3%	Histo and/or culture	Every 15 min for one h after ingestion of the urea solution	96.0	93.5
Wang et al. 1998 [37]	Taiwan	Patients underwent routine upper GI endoscopy	152	NM	NM	Cross-sectional	^13^C	100 mg	No	2.0	Histo and/or culture	30 min after ingestion	99.0	93.0
Van der Hulst et al. 1999 [38]	Italy	Patients with dyspeptic symptoms	199	48	105:94	Cross-sectional	^13^C	NM	Yes	>5%	Histo and culture	30 min after administration	96.0	97.0
Chen et al. 2000 [39]	Japan	Patients underwent GI endoscopy	169	53.9	101:68	Cross-sectional	^13^C	NM	No	2.5%	Combined (Histo and serology)	20 min after normal respiration	100.0	96.0
Hahn et al. 2000 [40]	United States	Patients with dyspeptic symptoms underwent upper GI endoscopy	67	58.8	61:6	Cross-sectional	^13^C	NM	No	>2.3%	Combined (Histo, UBT and serology)	30 min after administration	95.0	97.0
Riepl et al. 2000 [41]	Austria	Patients underwent gastroscopy with biopsy	100	51.6	51:49	Cross-sectional	^13^C	100 mg	Yes	>4%	Combined three tests (Histo, UAT and culture)	NM	92.0	94.0
Peng et al. 2000 [42]	China	Patients with gastritis and dyspeptic symptoms underwent upper GI endoscopy	136	47.6	66:70	Prospective cross-sectional	^13^C	NM	No	4.8%	Culture or combined (Histo and RUT)	15 min after ingestion	93.8	89.1
Wong et al. 2000 [43]	China	Patients with dyspeptic symptoms underwent upper GI endoscopy	202	49	90:112	Cross-sectional	^13^C	75 mg	No	5.0%	Histo and/or culture	30 min after ingestion	96.5	97.7
Wong et al. 2001 (a) [44]	China	Patients underwent upper GI endoscopy for the investigation of dyspepsia or for follow-up after *H. pylori* eradication and/or ulcer healing	206	48.9	97:109	Cross-sectional	^13^C	50 mg	No	3.0%	Histo and/or culture	30 min after ingestion	96.0	98.2
Wong et al. 2001 (b) [45]	China	Patients with dyspepsia underwent upper GI endoscopy	294	47.6	100:194	Cross-sectional	^13^C	75 mg	No	5.0%	Culture or combined (Histo and RUT)	30 min after ingestion	92.6	96.9
Gomes et al. 2002 [46]	Brazil	Patients underwent GI endoscopy	137	46.7	70:67	Cross-sectional	^14^C	185	No	1000–2000 CPM	Combined (Histo and RUT)	30 min post ingestion	95.0	98.3
Kato et al. 2002 [47]	Japan	Children underwent upper GI endoscopy	232	11.9	136:84	Cross-sectional	^13^C	100 mg	Yes	3.5%	Combined (Histology and rapid urease) and/or culture	20 min after ingestion	97.8	98.5
Kopański et al. 2002 [48]	Poland	Patients with chronic gastritis	92	45.5	56:36	Cross-sectional	^14^C	NM	No	>5%	Combined (Culture, serology, UBT and urine test for C-urea)	30 min after administration	100.0	89.5
Chua et al. 2002 [49]	Singapore	Patients underwent esophago-gastro-duodenoscopy	100	45	70:30	Prospective cross-sectional	^13^C	NM	No	NM	Histo and/or culture	30 min after ingestion	94.2	100.0
Chen et al. 2003 [50]	Taiwan	Patients underwent upper pan-endoscopy	586	45.7	306:280	Cross-sectional	^13^C	NM	Yes	≥2%	Culture alone or RUT	20 min after drinking solution	97.8	96.8
Gatta et al. 2003 [51]	Italy	Patients with dyspeptic symptoms	200	53	87:113	Prospective cross-sectional	^13^C	75	No	NM	Combined (Histology and rapid urease) and/or culture	30 min post ingestion	100.0	100.0
Gomollon et al. 2003 [52]	Spain	Patients underwent upper GI endoscopy	314	54.1	146:168	Prospective cross-sectional	^13^C	NM	No	≥5%	Culture and/or Combined (Histo and RUT)	30 min post ingestion	97.3	100.0
Valdepérez et al. 2003 [53]	Spain	Patients with dyspeptic symptoms	85	41.6	43:44	Cross-sectional	^13^C	NM	No	NM	Histo and RUT	30 min after administration	92.0	100.0
Wong et al. 2003 [54]	China	Patients underwent upper GI endoscopy	200	48.4	87:113	Cross-sectional	^13^C	100 mg	Yes	1.2%	Culture or combined (Histo and RUT)	30 min after ingestion	100.0	98.0
Öztürk et al. 2003 [55]	Turkey	Patients with dyspeptic symptoms	75	41	56:19	Cross-sectional	^14^C	37	No	100 DPM	Histology	NM	81.0	75.0
Urita et al. 2004 [56]	Japan	Patients with gastrointestinal symptoms underwent upper GI endoscopy	129	60.3	54:75	Cross-sectional	^13^C	NM	No	2.5%	Combined (Histology and rapid urease) and/or culture	30 min after ingestion	97.7	94.0
Gurbuz et al. 2005 [57]	Turkey	Patients with dyspeptic symptoms underwent upper GI endoscopy	65	42.4	22:46	Cross-sectional	^14^C	37	No	>50 CPM	Combined tests (Histo and RUT)	10 min after drinking solution	89.7	77.8
Peng et al. 2005 [58]	China	Patients underwent routine upper GI endoscopy	100	51.5	57:43	Cross-sectional	^13^C	NM	No	5%	Culture or combined (Histo and RUT)	30 min after ingestion	100.0	100.0
Ortiz-Olvera et al. 2007 [59]	Mexico	Patients underwent gastroscopy with biopsy	88	45	49:39	Cross-sectional	^13^C	NM	No	>4.22%	Culture and/or combined (Histo and RUT)	30 min after administration	90.2	93.3
Rasool et al. 2007 [60]	Pakistan	Patients with dyspeptic symptoms underwent upper GI endoscopy	94	40.8	60:34	Cross-sectional	^14^C	NM	No	>50 CPM	Two reference tests. Patient did both separately: (1) Histo; (2) RUT	After 10 min	92.0	93.0
Özdemir et al. 2008 [61]	Turkey	Patients with dyspeptic symptoms underwent upper GI endoscopy	89	45	30:59	Cross-sectional	^14^C	NM	No	25 < CPM as Heliprobe	Combined; any 2 positive (RUT, PCR and histo)	10 min after drinking solution	89.8	100.0
Calvet et al. 2009 [62]	Spain	Patients with dyspeptic symptoms	199	48.2	92:107	Cross-sectional	^13^C	NM	Yes	8.5%	Any two positive (Histopathology, RUT, UBT, and fecal serology)	20 min after drinking solution	99.2	60.5
Peng et al. 2009 [63]	Taiwan	Patients underwent upper GI endoscopy	100	55	56:44	Cross-sectional	^13^C	NM	Yes	4.8%	Culture or combined (Histo and RUT)	15 min after drinking solution	100.0	95.7
Bruden et al. 2011 [64]	Estonia	Patients underwent esophagogastroduodenoscopy	280	48	95:185	Cross-sectional	^13^C	NM	No	≥5%	Culture or (Histo and RUT)	NM	93.0	88.0
Wardi et al. 2012 [65]	Israel	Patients with partial gastrectomy underwent upper GI endoscopy	76	69.9	61:15	Retrospective cross-sectional	^13^C	NM	No	5%	Combined (Histology and rapid urease) and/or culture	Within 10 to 15 minutes	64.0	91.0
Sahni et al. 2015 [66]	India	Patients with dyspeptic symptoms underwent upper GI endoscopy	50	NM	NM	Cross-sectional	^14^C	NM	No	NM	Histo and/or culture	20 min after administration	88.0	83.0
Li et al. 2017 [67]	China	Patients with gastric cancer	21,857	45.6	9360:12,497	Cross-sectional	^13^C	NM	No	≥2.0%	Combined (Histology and rapid urease) and/or culture	30 min after administration	94.5	92.9
El-Shabrawi et al. 2018 [68]	Egypt	Children with dyspeptic symptoms	60	7.2	30:30	Prospective cross-sectional	^13^C	NM	No	>4.0%	Histo and/or culture	30 min after after ingestion	89.5	95.5
Kwon et al. 2019 [69]	South Korea	Patients with proven *H. pylori* infection undergone upper GI endoscopy	562	56.3	280:282	Prospective cross-sectional	^13^C	NM	Yes	2.5%	Combined (Histology and rapid urease) and/or culture	20 minutes after administration	91.7	81.3
Alzoubi et al. 2020 [70]	Jordan	Patients with gastrointestinal diseases symptoms underwent upper GI endoscopy	60	37.3	24:36	Cross-sectional	^13^C	NM	Yes	NM	Histo and/or culture	30 min after ingestion	94.1	76.9
Miftahussurur et al. 2021 [71]	Indonesia	Patients with dyspeptic symptoms underwent upper GI endoscopy	55	19	36:19	Cross-sectional	^14^C	NM	No	57	Combined (Histology and rapid urease) and/or culture	10 min after ingestion	92.3	97.6
Hussein et al. 2021 [72]	Iraq	Patients with gastrointestinal diseases symptoms underwent upper GI endoscopy	115	40.3	80:35	Cross-sectional	^14^C	NM	No	NM	Histo and/or culture	10 min after ingestion	97.5	97.0

**Table 2 diagnostics-12-02428-t002:** Effect analysis of included articles.

Model	Effect Size and 95% Interval				Test of Null (2-Tail)		Heterogeneity				Tau-Squared			
Model	Number of Studies	Point of Estimate	Lower Limit	Upper Limit	Z-Value	*p*-Value	Q-Value	df (Q)	*p*-Value	I-Squared	Tau Squared	Standard Error	Variance	Tau
Sensitivity														
**Fixed**	41	0.925	0.922	0.929	97.195	0.000	2137.239	40	0.000	98.128	3.350	2214	4.902	1.830
**Random**	41	0.876	0.799	0.926	6.681	0.000								
**Specificity**														
**Fixed**	41	0.899	0.895	0.903	95.535	0.000	2689.815	40	0.000	98.516	3.272	2.239	5.014	1.809
**Random**	41	0.848	0.760	0.908	5.937	0.000								

**Table 3 diagnostics-12-02428-t003:** Classic and Orwin’s Fail-Safe N findings.

Sensitivity				Specificity			
Classic Fail-Safe N Method		Orwin’s Fail-Safe N Method		Classic Fail-Safe N Method		Orwin’s Fail-Safe N Method	
Z-value for observed studies	44.48086	The event rate is observed in studies	0.92518	Z-value for observed studies	40.05841	The event rate is observed in studies	0.89914
The *p*-value for observed studies	0.00000	The criterion for a “trivial” event rate	0.50000	The *p*-value for observed studies	0.00000	The criterion for a “trivial” event rate	0.50000
Alpha	0.05000	Mean event rate in missing studies	0.50000	Alpha	0.05000	Mean event rate in missing studies	0.50000
Tails	2.00000	Number of missing studies that would bring the *p*-value to > alpha (N value)	The criterion must fain between other values	Tails	2.00000		
Z for alphas	1.95996			Z for alphas	1.95996	Number of missing studies that would bring the *p*-value to > alpha (N value)	The criterion must fain between other values
Number of observed studies	41.00000			Number of observed studies	41.00000		
Number of missing studies that would bring the *p*-value to > alpha (N value)	1077.00000			Number of missing studies that would bring the *p*-value to > alpha (N value)	7086.00000		

**Table 4 diagnostics-12-02428-t004:** Kendall’s tau with/without continuity correction and Egger’s regression intercept.

	Sensitivity	Specificity
Kendall’s S Statistic (P-Q)	236.000	325.00000
Kendall’s tau with continuity correction		
Tau	0.28729	0.39560
z-value for tau	2.63951	3.63915
*p*-value (1-tailed)	0.00415	0.00014
*p*-value (2-tailed)	0.900830	0.00027
**Kendall’s tau without continuity correction**		
Tau	0.28851	0.39683
z-value for tau	2.65074	3.65038
*p*-value (1-tailed)	0.00402	0.00013
*p*-value (2-tailed)	0.00803	0.00026
**Egger’s regression intercept**		
Intercept	−3.12604	−3.39677
Standard error	1.34367	1.48526
95% low limit (2-tailed)	−5.84387	−6.40099
95% upper limit (2-tailed)	−0.40820	−0.39254
t-value	2.32648	2.28698
df	39.00000	39.00000
*p*-value (1-tailed)	0.01264	0.01385
*p*-value (2-tailed)	0.02528	0.02771

**Table 5 diagnostics-12-02428-t005:** Subgroup analysis.

Subgroup	Number of Articles	Sensitivity Effect Size and 95% Confidence Interval	Specificity Effect Size and 95% Confidence Interval
**^13^C-Urea Breath Tests**	29	0.791 (0.742–0.825)	0.750 (0.700–0.786)
**^14^C-Urea Breath Tests**	12	0.780 (0.702–0.831)	0.766 (0.685–0.816)

## Data Availability

Not applicable.

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
