# Peer review of "Systematic Review and Meta-Analysis on the Sensitivity and Specificity of 13C/14C-Urea Breath Tests in the Diagnosis of Helicobacter pylori Infection"

_diagnostics, 2022, doi:10.3390/diagnostics12102428_

Round 1

Reviewer 1 Report

In the present meta analysis Jambi showed that the pooled sensitivity and specificity of urea breath test (UBT) for H. pylori diagnosis were close or higher than 90% and, therefore, optimal for diagnosis. Main comments:

1) H. pylori should be italicized

2) Page 1 line 39: “insanitary nations”?

3) Page 1 lines 41-43: please check the sentence

4) Lines 94, 96: helicobacter - - > Helicobacter

5) The fact that the meta-analysis was conducted by only one Author is a relevant issue since a second or a third reviewer is frequently required to avoid errors in study selection, data extraction, or to solve inconsistencies.

6) Please calculate also positive and negative likelihood ration and diagnostic odd ratio.

7) The high heterogeneity should have been solved by performing subgroup analysis. For example, it is important to calculate sensitivity and specificity separately for 13C and 14C.

8) Please explain the meaning of rank correlation test (paragraph 3.6).

9) Several other similar meta-analyses have been already published on the topic (see n. 21 in the reference list). Please discuss the novelties of the present one.

Author Response

September 16, 2022

Diagnostics

MDPI publisher

Manuscript ID: diagnostics-1922714

Resubmission of manuscript "Systematic Review and Meta-Analysis on the Sensitivity and Specificity of 13C/14C-Urea Breath Tests in the Diagnosis of Helicobacter Pylori Infection ".

Ms. Estella Shu

Assistant Editor

Diagnostics

MDPI publisher

Thank you for the opportunity to revise our manuscript. We appreciate the careful review and constructive suggestions. It is our belief that the manuscript is substantially improved after making the suggested edits. Following this letter are the reviewers' comments with our responses in red, including how and where the text was modified. The revision has been developed in consultation with all coauthors, and each author has given approval to the final form of this revision.

Sincerely,

Dr. Layal Jambi

Response to Reviewer(s)' Comments:

Reviewer #1:

Comments and Suggestions for Authors

In the present meta analysis Jambi showed that the pooled sensitivity and specificity of urea breath test (UBT) for H. pylori diagnosis were close or higher than 90% and, therefore, optimal for diagnosis. Main comments:

  • pylori should be italicized

- Done.

2) Page 1 line 39: “insanitary nations”?

- We have rewritten this sentence as following: In addition, in countries with poor sanitation, 90% of the adult population can be affected [8].

3) Page 1 lines 41-43: please check the sentence

- Thank you for this important comment. We have rewritten the paragraph as “Most H. pylori infected people never experience any signs of the infection, whereas H. pylori causes chronic active, chronic persistent, and atrophic gastritis in both adults and children [9]. Furthermore, contamination with H. pylori causes gastric and duodenal ulcers [10]”.

4) Lines 94, 96: helicobacter - - > Helicobacter

- Corrected.

5) The fact that the meta-analysis was conducted by only one Author is a relevant issue since a second or a third reviewer is frequently required to avoid errors in study selection, data extraction, or to solve inconsistencies.

- Thank you for this vital notice, we have added the following sentence to the acknowledgement section “The author is grateful to Dr Samer Abuzerr and Dr Saeed M. Kabrah for their help in some data analysis, article screening, data extraction, and reviewing the manuscript.”

6) Please calculate also positive and negative likelihood ration and diagnostic odd ratio.

- Please note that this meta-analysis was based on event rate, accordingly, calculating diagnostic odds ratio is not applicable.

7) The high heterogeneity should have been solved by performing subgroup analysis. For example, it is important to calculate sensitivity and specificity separately for 13C and 14C.

- Thank you for raising this insightful note. A new table presenting the subgroup analysis was added to the manuscript to calculate the sensitivity and specificity separately for 13C and 14C. However, please note that the observed heterogeneity can be attributed to several factors including (sample size, study design, country of origin, etc.), not only 13C and 14C subgroups.

8) Please explain the meaning of rank correlation test (paragraph 3.6).

- The rank correlation test introduced by Begg and Mazumdar is extensively used in meta-analysis to test for publication bias in clinical and epidemiological studies. It is based on correlating the standardized treatment effect with the variance of the treatment effect using Kendall's tau as the measure of association.

9) Several other similar meta-analyses have been already published on the topic (see n. 21 in the reference list). Please discuss the novelties of the present one.

- Thank you for raising this important comment. Ferwana mata-analysis was conducted in 2015 and included 23 articles, while ours was conducted 2022 and included 41 articles. We only focused on the sensitivity and specificity of 13C/14C-urea breath tests in the diagnosis of helicobacter pylori infection. We further discussed the novelties of the present meta-analysis in the manuscript strength section.

Reviewer 2 Report

diagnostics-1922714-peer-review-v1

This manuscript evaluates the sensitivity and specificity of the urea breath test for Helicobacter pylori infection by meta-analysis. It can be seen that the authors carefully and deliberately selected papers suitable for this analysis using quantitative indices to avoid arbitrary selection of papers.

It would also be of material value to list the methods of urea breath testing that have been independently performed at numerous institutions and show how the sensitivity and specificity have changed.

On the other hand, the presentation of the problems that motivate this analysis and the discussion based on the results are insufficient.

L70-72

In actuality, it is unknown whether non-invasive tests are more reliable for detecting H. pylori infection, and it is also unclear what clinical value the levels discovered by these tests have [22].

Although the authors question the validity of such noninvasive testing, in the conclusion of the Best et al. paper, which the authors refer to as [22], it is stated that

"In people without a history of gastrectomy and those who have not recently had antibiotics or proton ,pump inhibitors, urea breath tests In people without a history of gastrectomy and those who have not recently had antibiotics or proton ,pump inhibitors, urea breath tests had high diagnostic accuracy while serology and stool antigen tests were less accurate for diagnosis of Helicobacter pylori infection. This is based on an indirect test comparison (with potential for bias due to confounding), as evidence from direct comparisons was limited or unavailable. The thresholds used for these tests were highly variable and we were unable to identify specific thresholds that might be useful in clinical practice. We need further comparative studies of high methodological quality to obtain more reliable evidence of relative accuracy between the tests. Such studies should be conducted prospectively in a representative spectrum of participants and clearly reported to ensure low risk of bias, Most importantly, studies should prespecify and clearly report thresholds used, and should avoid inappropriate exclusions."

Thus, there are no negative statements regarding the validity of the urea breath test and the clinical significance of the test results cited by the authors. It is doubtful whether the authors' statements are appropriate as their paraphrase.

In addition, if this section is absent from the manuscript, there will be a lack of reason why this study was necessary, and a new reference should be provided that clearly points out that the urea breath test results are unreliable in terms of their specificity and sensitivity.

The discussion also lacks a description of new findings related to the urea breath test. The heterogeneity across studies and the superior sensitivity and specificity of the urea breath test are not particularly new findings.

If you want to consider heterogeneity as a new finding, you should analyze and mention, for example, whether the heterogeneity stood out compared to other bacterial detection tests or other tests for H. pylori. If this would result in a finding that "the protocol of this test is too disparate compared to other testing methods," then that would be a new finding and raise new issues that would make this manuscript useful and valuable.

Or you should point out some interesting phenomenon, such as considering how this examination has changed over time.

All words of H. pylori should be italic.

Author Response

September 16, 2022

Diagnostics

MDPI publisher

Manuscript ID: diagnostics-1922714

Resubmission of manuscript "Systematic Review and Meta-Analysis on the Sensitivity and Specificity of 13C/14C-Urea Breath Tests in the Diagnosis of Helicobacter Pylori Infection ".

Ms. Estella Shu

Assistant Editor

Diagnostics

MDPI publisher

Thank you for the opportunity to revise our manuscript. We appreciate the careful review and constructive suggestions. It is our belief that the manuscript is substantially improved after making the suggested edits. Following this letter are the reviewers' comments with our responses in red, including how and where the text was modified. The revision has been developed in consultation with all coauthors, and each author has given approval to the final form of this revision.

Sincerely,

Dr. Layal Jambi

Response to Reviewer(s)' Comments:

Reviewer #2:

Comments and Suggestions for Authors

This manuscript evaluates the sensitivity and specificity of the urea breath test for Helicobacter pylori infection by meta-analysis. It can be seen that the authors carefully and deliberately selected papers suitable for this analysis using quantitative indices to avoid arbitrary selection of papers.

It would also be of material value to list the methods of urea breath testing that have been independently performed at numerous institutions and show how the sensitivity and specificity have changed.

  • Thank you for raising this important point. We have further discussed it in the introduction section.

 On the other hand, the presentation of the problems that motivate this analysis and the discussion based on the results are insufficient.

In the discussion section, we further discussed the motivation of this study.

L70-72

“In actuality, it is unknown whether non-invasive tests are more reliable for detecting H. pylori infection, and it is also unclear what clinical value the levels discovered by these tests have [22].”

Although the authors question the validity of such noninvasive testing, in the conclusion of the Best et al. paper, which the authors refer to as [22], it is stated that

 "In people without a history of gastrectomy and those who have not recently had antibiotics or proton ,pump inhibitors, urea breath tests In people without a history of gastrectomy and those who have not recently had antibiotics or proton ,pump inhibitors, urea breath tests had high diagnostic accuracy while serology and stool antigen tests were less accurate for diagnosis of Helicobacter pylori infection. This is based on an indirect test comparison (with potential for bias due to confounding), as evidence from direct comparisons was limited or unavailable. The thresholds used for these tests were highly variable and we were unable to identify specific thresholds that might be useful in clinical practice. We need further comparative studies of high methodological quality to obtain more reliable evidence of relative accuracy between the tests. Such studies should be conducted prospectively in a representative spectrum of participants and clearly reported to ensure low risk of bias, Most importantly, studies should prespecify and clearly report thresholds used, and should avoid inappropriate exclusions."

 Thus, there are no negative statements regarding the validity of the urea breath test and the clinical significance of the test results cited by the authors. It is doubtful whether the authors' statements are appropriate as their paraphrase.

 In addition, if this section is absent from the manuscript, there will be a lack of reason why this study was necessary, and a new reference should be provided that clearly points out that the urea breath test results are unreliable in terms of their specificity and sensitivity.

  • Thank you for this important comment. We have further discussed these points in the discussion section.

The discussion also lacks a description of new findings related to the urea breath test.

  • Further discussion was added to cover this important point.

The heterogeneity across studies and the superior sensitivity and specificity of the urea breath test are not particularly new findings.

If you want to consider heterogeneity as a new finding, you should analyze and mention, for example, whether the heterogeneity stood out compared to other bacterial detection tests or other tests for H. pylori. If this would result in a finding that "the protocol of this test is too disparate compared to other testing methods," then that would be a new finding and raise new issues that would make this manuscript useful and valuable.

Or you should point out some interesting phenomenon, such as considering how this examination has changed over time.

  • Thank you for this insightful comment. You are opening mind to do a new systematic review and meta-analysis to identify whether the heterogeneity stood out compared to other bacterial detection tests or other tests for H. pylori. We promise to take it in consideration in future research. However, a new table presenting the subgroup analysis was added to the manuscript to calculate the sensitivity and specificity separately for 13C and 14 However, please note that the observed heterogeneity can be attributed to several factors including (sample size, study design, country of origin, etc.), not only 13C and 14C subgroups.

All words of H. pylori should be italic.

  • Done.

Round 2

Reviewer 1 Report

Regarding point 6, I do not agree. Indeed other softwares, such as MetaDisc, are able to calculate positive and negative predictive value and S-ROC, which are absent in this analysis.

All other answers were perfect.

Author Response

September 22, 2022

Diagnostics

MDPI publisher

Manuscript ID: diagnostics- 1922714

Resubmission of manuscript "Systematic Review and Meta-Analysis on the Sensitivity and Specificity of 13C/14C-Urea Breath Tests in the Diagnosis of Helicobacter Pylori Infection ".

Ms. Estella Shu

Assistant Editor

Diagnostics

MDPI publisher

Thank you for the opportunity to revise our manuscript. We appreciate the careful review and constructive suggestions. It is our belief that the manuscript is substantially improved after making the suggested edits. Following this letter are the reviewers' comments with our responses in red, including how and where the text was modified. The revision has been developed in consultation with all coauthors, and each author has given approval to the final form of this revision.

Sincerely,

Dr. Layal Jambi

Response to Reviewer(s)' Comments:

Reviewer #1:

Comments and Suggestions for Authors

Regarding point 6, I do not agree. Indeed other softwares, such as MetaDisc, are able to calculate positive and negative predictive value and S-ROC, which are absent in this analysis.

  • Done.

Reviewer 2 Report

L81-83

The problems noted have not been remedied.

The paper cited by the author states in its conclusion that the urea breath test has a high diagnostic accuracy compared to fecal antigen tests and serological tests.

The paper also clearly states that there was no significant difference in the sensitivity and specificity of the urea breath test between using 13C and 14C.

It is inappropriate to cite the paper in [25] as the basis for the text in L 81-83.

Either cite another paper that follows the text in L 81-83, or delete this sentence and provide another sentence explaining the motivation for conducting this study.

However, in the paper [25], as a look ahead to the future, we need further comparative studies of high methodological quality to obtain more reliable evidence of relative accuracy between the tests. Such studies should be conducted prospectively in a representative spectrum of participants and clearly reported to ensure low risk of bias. Most importantly, studies should prespecify and clearly report thresholds used, and should avoid inappropriate exclusions. If this manuscript corresponds to this "further comparative studies", it would be a good idea to clearly state so and cite the article in [25].

In any case, the text in L 81-83 is unacceptable because it says the opposite of what is quoted.

L 259-271

I did not point out that the conclusions of the Best et al. paper should be included in the discussion. It should be revised as first stated in this report, and this section is an unnecessary and inappropriate citation and should be deleted.

L 287-295, 314-316

I think this part of the article corresponds to the new finding I pointed out, but if this is a new finding, the conclusion must also be changed.

L319-320, "For the diagnosis of H. pylori infection, the 13C/14C-urea breath tests are extremely sensitive and specific." This is a known fact, as stated.

Please reflect the new findings in your conclusion.

Please check again that H. pylori and Helicobacter pylori are all in italics.

Author Response

September 22, 2022

Diagnostics

MDPI publisher

Manuscript ID: diagnostics-1922714

Resubmission of manuscript "Systematic Review and Meta-Analysis on the Sensitivity and Specificity of 13C/14C-Urea Breath Tests in the Diagnosis of Helicobacter Pylori Infection ".

Ms. Estella Shu

Assistant Editor

Diagnostics

MDPI publisher

Thank you for the opportunity to revise our manuscript. We appreciate the careful review and constructive suggestions. It is our belief that the manuscript is substantially improved after making the suggested edits. Following this letter are the reviewers' comments with our responses in red, including how and where the text was modified. The revision has been developed in consultation with all coauthors, and each author has given approval to the final form of this revision.

Sincerely,

Dr. Layal Jambi

Response to Reviewer(s)' Comments:

Reviewer #2:

Comments and Suggestions for Authors

L81-83

The problems noted have not been remedied.

The paper cited by the author states in its conclusion that the urea breath test has a high diagnostic accuracy compared to fecal antigen tests and serological tests.

The paper also clearly states that there was no significant difference in the sensitivity and specificity of the urea breath test between using 13C and 14C.

It is inappropriate to cite the paper in [25] as the basis for the text in L 81-83.

Either cite another paper that follows the text in L 81-83, or delete this sentence and provide another sentence explaining the motivation for conducting this study.

 However, in the paper [25], as a look ahead to the future, we need further comparative studies of high methodological quality to obtain more reliable evidence of relative accuracy between the tests. Such studies should be conducted prospectively in a representative spectrum of participants and clearly reported to ensure low risk of bias. Most importantly, studies should prespecify and clearly report thresholds used, and should avoid inappropriate exclusions. If this manuscript corresponds to this "further comparative studies", it would be a good idea to clearly state so and cite the article in [25].

 In any case, the text in L 81-83 is unacceptable because it says the opposite of what is quoted.

Omitted

 L 259-271

I did not point out that the conclusions of the Best et al. paper should be included in the discussion. It should be revised as first stated in this report, and this section is an unnecessary and inappropriate citation and should be deleted.

Deleted.

 L 287-295, 314-316

I think this part of the article corresponds to the new finding I pointed out, but if this is a new finding, the conclusion must also be changed.

L319-320, "For the diagnosis of H. pylori infection, the 13C/14C-urea breath tests are extremely sensitive and specific." This is a known fact, as stated.

Please reflect the new findings in your conclusion.

We have changed the conclusion based on the new finding.

Please check again that H. pylori and Helicobacter pylori are all in italics.

Done.

Round 3

Reviewer 1 Report

Answers were fine.

Author Response

Dear respected reviewer,

Thank you for the opportunity to revise our manuscript. We appreciate the careful review and constructive suggestions. It is our belief that the manuscript is substantially improved after making the suggested edits.

Response to Reviewer(s)' Comments:

Reviewer #1:        

Answers were fine.

Reviewer 2 Report

Major rivision

As I pointed out in the first peer review report, if you simply delete this sentence, the motivation for conducting this study cannot be read from the instructions.

In place of the deleted sentence, please provide the sentence and its references that present problems with the urea breath test or its threshold.

Do not list the deleted paper of Best et al again. That was not the paper that questioned the clinical evaluation of the urea breath test itself.

If only this point can be resolved, I will accept it.

Minor rivision

Some of the newly added letters for H. pylori are not italicized.

The numbers 13 and 14, which indicate isotopes, are sometimes superscripted and sometimes not. It would be better to unify them.

Author Response

Dear respected reviewer,
Thank you for the opportunity to revise our manuscript. We appreciate the careful review and constructive suggestions. It is our belief that the manuscript is substantially improved after making the suggested edits. 

Response to Reviewer(s)' Comments:
Reviewer #2:
Minor revision
Some of the newly added letters for H. pylori are not italicized.
- Corrected, all H. pylori are italicized now.
The numbers 13 and 14, which indicate isotopes, are sometimes superscripted and sometimes not. It would be better to unify them.
- Corrected, all are superscripted.

Major rivision

As I pointed out in the first peer review report, if you simply delete this sentence, the motivation for conducting this study cannot be read from the instructions.

In place of the deleted sentence, please provide the sentence and its references that present problems with the urea breath test or its threshold.

Do not list the deleted paper of Best et al again. That was not the paper that questioned the clinical evaluation of the urea breath test itself.

If only this point can be resolved, I will accept it.

 - I have added further information to the introduction section about the problems with the urea breath test and justifying the importance of the attached current systematic review and meta-analysis citing three new references (written in red).
